# In Vivo Neuropharmacological Effects of Neophytadiene

**DOI:** 10.3390/molecules28083457

**Published:** 2023-04-14

**Authors:** Maria L. Gonzalez-Rivera, Juan Carlos Barragan-Galvez, Deisy Gasca-Martínez, Sergio Hidalgo-Figueroa, Mario Isiordia-Espinoza, Angel Josabad Alonso-Castro

**Affiliations:** 1Departamento de Farmacia, División de Ciencias Naturales y Exactas, Universidad de Guanajuato, Guanajuato 36200, Mexico; 2Unidad de Análisis Conductual, Instituto de Neurobiología, Campus UNAM-Juriquilla, Juriquilla 76230, Mexico; 3CONACyT-División de Biología Molecular, Instituto Potosino de Investigación Científica y Tecnológica A.C., San Luis Potosí 78216, Mexico; 4Instituto de Investigación en Ciencias Médicas, Departamento de Clínicas, División de Ciencias Biomédicas, Centro Universitario de los Altos, Universidad de Guadalajara, Tepatitlán de Morelos 47620, Mexico

**Keywords:** neophytadiene, anxiety, convulsion, diterpene

## Abstract

Neophytadiene (NPT) is a diterpene found in the methanolic extracts of *Crataeva nurvala* and *Blumea lacera*, plants reported with anxiolytic-like activity, sedative properties, and antidepressant-like actions; however, the contribution of neophytadiene to these effects is unknown. This study determined the neuropharmacological (anxiolytic-like, antidepressant-like, anticonvulsant, and sedative) effects of neophytadiene (0.1–10 mg/kg p.o.) and determined the mechanisms of action involved in the neuropharmacological actions using inhibitors such as flumazenil and analyzing the possible interaction of neophytadiene with GABA receptors using a molecular docking study. The behavioral tests were evaluated using the light–dark box, elevated plus-maze, open field, hole-board, convulsion, tail suspension, pentobarbital-induced sleeping, and rotarod. The results showed that neophytadiene exhibited anxiolytic-like activity only to the high dose (10 mg/kg) in the elevated plus-maze and hole-board tests, and anticonvulsant actions in the 4-aminopyridine and pentylenetetrazole-induced seizures test. The anxiolytic-like and anticonvulsant effects of neophytadiene were abolished with the pre-treatment with 2 mg/kg flumazenil. In addition, neophytadiene showed low antidepressant effects (about 3-fold lower) compared to fluoxetine. On other hand, neophytadiene had no sedative or locomotor effects. In conclusion, neophytadiene exerts anxiolytic-like and anticonvulsant activities with the probable participation of the GABAergic system.

## 1. Introduction

In the world, neurological disorders are the leading cause of disability and the second leading cause of death (8.8 million people died in 2019). Only in the USA and Europe, the annual cost of treating mental disorders is USD 1.7 trillion [1]. More than 50 million and 322 million individuals suffer from epilepsy and depression, respectively [2]. Multiple investigations have indicated that anxiety and depression might simultaneously occur [3]. Scientific research has focused on studying medicinal plants and the discovery of potentially bioactive compounds for treating neurological disorders [4]. Several potential compounds isolated from plant extracts have been assayed as possible treatments for anxiety, depression, and epilepsy [4]. For instance, methanolic extracts from the leaves of *Blumea lacera* and *Crataeva nurvala* showed anxiolytic-like, sedative, and antidepressant activities with inhibition of locomotor activity in mice. Several bioactive metabolites were identified in these two studies, and one of the compounds was neophytadiene (NPT), a diterpene (Figure 1) [3,5]. Diterpenes with anticonvulsant effects remained to be analyzed, such as phytol, which increased the latency of onset clonic and tonic seizures and the survival rate with minimal motor impairment in a model of convulsion induced by pentylenetetrazole (PTZ) [4,6].

Medicinal plants containing NPT are used in the treatment of headaches, rheumatism, and some skin diseases, whereas NPT has shown analgesic, antipyretic, anti-inflammatory, and antioxidant properties [7,8,9,10,11,12]. The anti-inflammatory effect of NPT is attributed to reducing nitric oxide (NO) production and decreasing mRNA levels of proinflammatory cytokines (IL-6 and IL-10) and inhibiting the production of tumor necrosis factor α (TNFα), nuclear factor kB (NF-kB), and downregulated production of inducible nitric synthase (iNOS) in an in vitro assay [13]. NPT also showed antimicrobial activity against gram-positive and gram-negative bacteria, and fungal species [7,14,15].

Few studies have evaluated the pharmacological effects of NPT. To our knowledge, no scientific research has focused the analysis of NPT on mental disorders and seizure protection. Therefore, this research focused on evaluating the anxiolytic-like, antidepressant-like, anticonvulsant, and sedative effects of NPT using in vivo models.

## 2. Results

### 2.1. Anxiolytic-like Activity of NPT and Its Possible Mechanism of Action 

In the light–dark box test (LDBT), the three tested doses of NPT did not show statistical significance (*p* > 0.05) in the time and the number of entries in the light compartment compared to the vehicle group, whereas the reference drug clonazepam (CNZ) raised (*p* < 0.05) the time and the number of entries in the light compartment (Figure 2a,b).

In the elevated plus-maze test (EPMT), only the high dose of NPT (10 mg/kg), compared to the vehicle group, increased (*p* < 0.05) the time in open arms and decreased (*p* < 0.05) the number of entries and time in closed arms (Figure 3b,d); however, this same dose did not affect the number of entries in open arms (Figure 3a). The anxiolytic-like effect presented by NPT was lower than that presented by CNZ.

In the open field test (OFT) (Table 1), no dose of NPT elicited anxiolytic-like activity compared to 1.5 mg/kg of the reference drug CNZ, which decreased the total distance and increased the resting time, time in the center squares, and distance in center squares (*p* < 0.05 vs. vehicle group).

In the hole-board test (HBT), only 10 mg/kg NPT increased (*p* < 0.05) the occurrence of head dipping by 2.6-fold compared to the vehicle group (Figure 4a). The anxiolytic-like activity of NPT was close to the effect shown by the reference drug CNZ (3.3-fold). On the other hand, Figure 4b shows that the pre-treatments with the antagonists of GABA receptor bicuculline (1 mg/kg) and flumazenil (2 mg/kg) abolished the anxiolytic-like effects shown by NPT, showing similar results to the vehicle group (*p* > 0.05). These findings suggest that 10 mg/kg of NPT shows an anxiolytic-like effect.

A molecular docking study evaluated the possible interaction of NPT in GABA_A_ receptors (Figure 5). NPT exhibits the same preference in binding site as flumazenil (Figure 5A–C). Interestingly, flumazenil and NPT can form hydrophobic interactions in aspartic acid residue at position 56 (Asp56), tyrosine residue at position 58 (Tyr58), phenylanalanine residue at position 77 (Phe77), alanine residue at position 79 (Ala79), histidine residue at position 102 (His102), threonine residue at position 142 (Thr142), tyrosine residue at position 210 (Tyr210), and tyrosine residue at position 260 (Tyr260) of GABA_A_ receptors (Figure 5B,C). Flumazenil showed an additional H-bond with Thr142. The binding energy for flumazenil (redocked) was −9.4 Kcal/mol, whereas the binding energy for NPT was −7.1 Kcal/mol.

### 2.2. Anticonvulsant Activity and Possible Mechanism of Action

NPT (0.1–10 mg/kg) decreased the duration of the convulsion and decreased the mortality rate in a dose-dependent compared to the vehicle group in the pentylenetetrazole (PTZ)-induced convulsion test (Table 2). It is important to mention that two mice treated with 10 mg/kg NPT did not show convulsions. The pretreatment with 2 mg/kg flumazenil abolished the anticonvulsant effects of 10 mg/kg NPT, increasing the mortality rate (Table 2, upper section). In the 4-aminopyridine (4-AP)-induced convulsion, the three doses of NPT decreased the mortality rate compared to the vehicle group; the dose of 10 mg/kg had a significant (*p* < 0.05) effect on the onset of convulsion and the 0.1 mg/kg and 1.0 mg/kg doses significantly decreased the duration of convulsion (Table 2, bottom section). This data shows that NPT exhibits anticonvulsant effects and reduces the mortality for seizures.

### 2.3. Antidepressant Effects

In the tail suspension test (TST), the immobility time of the groups treated with NPT decreased significantly (*p* < 0.05) by 22.3% (0.1 mg/kg), 17.7% (1 mg/kg), and 19.1% (10 mg/kg) compared to the vehicle group, but this antidepressant-like activity was lower compared to the group treated with 20 mg/kg of the reference drug FLX (66.2%) (Figure 6). Therefore, no further antidepressant-like assays were performed.

### 2.4. Sedative Effects and Locomotor Actions 

The mice treated with NPT showed no significant (*p* > 0.05) difference in the beginning of sleep and length of sleep in the model of sedation produced with pentobarbital, showing the same results with the vehicle group and opposite effects to the group treated with the reference drug CNZ (Figure 7). 

For the rotarod test (RT), the three doses of NPT did not affect the motor coordination of the mice compared to 5 mg/kg CNZ (reference drug), which decreased the time in the rotarod by 37.4% (60 min) and 57.5% (120 min) (Figure 8). 

## 3. Discussion

There are many plants and plant-derived compounds used for treating different ailments, but their pharmacological effects are still unknown. New treatments for mental disorders are necessary due to these affections decrease the quality of life, increase the cost of stays in the hospital, and diminish productivity at work [16]. For this reason, this study evaluated the neuropharmacological activity of NPT and provided evidence of anxiolytic-like and anticonvulsant actions of this diterpene.

The behavioral tests used for anxiety evaluation were LDBT, EPMT, OFT, and HBT. These assays are based on the conflict that each animal experiences between exploring a new environment and rodents’ natural aversion to bright, elevated, and open spaces [17]. CNZ was used as a positive control due to the anxiolytic activity demonstrated in clinical increase trials [18]. CNZ showed in all experimental tests related to anxiety that the mice overcame the aversion to bright and new, open spaces. CNZ belongs to the benzodiazepine family and is a positive allosteric modulator on GABA_A_ receptors that hyperpolarizes neurons by increasing the opening of Cl^-^ channel, reducing the excitability of neurons [19].

Only 10 mg/kg NPT presented anxiolytic activity in 2 (EPMT and HBT) of the 4 models of anxiety in mice. In the EPMT, NPT increased the rodent’s time elapsed in an unprotected area (open arms) and decreased the number of entries and the time in a protected place (closed arms); however, these effects were not comparable to those shown by CNZ. In the HBT, NPT increased the number of head dips in the hole board with an anxiolytic activity such as that elicited by CNZ. Other diterpenes such as 3,4-secoisopimar-4(18),7,15-triene-3-oic acid (CMP1) and rosmanol, intraperitoneally administered and tested at similar doses as NPT, both showed anxiolytic-like activities by increasing the time and the number of entries in open arms in the EPMT [20,21]. In addition, other diterpenes, such as tilifodiolide (TFD), tested at 50 and 100 mg/kg p.o., and neo-clerodane 7-keto-neoclerodan-3,13-dien-18,19:15,16-diolide, tested at 10 mg/kg p.o., both showed anxiolytic-like effects in the EPMT [22,23].

Furthermore, the anxiolytic-like mechanism of NPT (10 mg/kg) was evaluated using the GABA inhibitors flumazenil and bicuculline in the HBT. The results showed that NPT effects are reversed in the individual co-administrations of these drugs. These actions were due to the antagonism produced by flumazenil and bicuculline on the GABA_A_ receptors [24,25,26]. Flumazenil is a potent benzodiazepine receptor antagonist that competitively inhibits the activity of the benzodiazepine recognition site of the subunits α_2_ and α_3_ of the GABA_A_ receptor implicated in anxiolytic-like and anticonvulsant effects [27]. Based on the molecular docking study, the hypothesis is that neophytadiene has the potential to occupy the benzodiazepine site and exerts anxiolytic-like and anticonvulsant effects through the GABAergic system without inducing impairments in motor coordination or sedation. According to random molecular docking, this study found that NPT has an affinity for the same flumazenil binding site and not for other available GABA_A_ receptor binding sites. 

Furthermore, these findings would explain the anticonvulsant effects of NPT presented in the convulsion models induced with PTZ, where 10 mg/kg NPT decreased the duration of convulsion and the mortality rate by at least 50%. The convulsion mechanism for 4-AP is induced by selectively blocking voltage-activated K^+^ channel [28] where the anticonvulsant action of NPT was different, only decreasing seizure-related mortality 15%, but interestingly, NPT showed a partial anticonvulsant effect in onset and duration of convulsion. A possible theory is that NPT could also interact on K^+^ channels but with a lower affinity than seen for GABA_A_ receptors. 

The mechanism of action found by NPT has not been the same as other diterpenes. For instance, the anxiolytic-like effects of TFD and rosmanol failed or were slightly reversed with the same inhibitor used in this study (flumazenil), demonstrating that the anxiolytic-like activity of TFD and rosmanol is probably not related to the participation of the GABAergic system [21,22].

Few diterpenes with anticonvulsant effects have been reported. For instance, diterpene lactone reported in leaf extracts from *Leonotis leonorus* showed protection against PTZ-induced seizures by up to 50% at the high dose of 400 mg/kg (i.p.); however, the mechanism of the anticonvulsant activity was not elucidated [29]. The diterpene phytol, at the high dose of 250 mg/kg (i.p.), suppressed the PTZ-induced convulsions with minimal locomotor impairments. The anticonvulsant activity was abolished with the coadministration of flumazenil, demonstrating the possible participation of the GABAergic system [6]. In this study, 10 mg/kg p.o. NPT protected mice and delayed the onset of convulsions in the PTZ-induced convulsion test. As shown in the docking study, GABAergic participation is the possible mechanism of the anticonvulsant activity shown by NPT.

The tail suspension test is a model consisting in recreating a despair behavior reflected by a failure of mice in the persistent attempt to escape from an inescapable situation [30]. NPT (0.1, 1, and 10 mg/kg p.o.) moderately reduced the time the mice remained immobile, indicating a low antidepressant-like effect (17.7–22.3%), which was the same regardless of the increase in its dose, whereas 20 mg/kg fluoxetine showed an antidepressant effect of 66.2%, being nearly 3-fold higher than that of NPT. These data indicate that NPT would be a compound with low effectiveness for treating depression, but it could be synergized in combination with other diterpenes, such as the diterpene TFD, which is reported to have a better antidepressant effect at doses of 10, 50, and 100 mg/kg p.o. [22]. 

In neurological studies, diterpenes such as CMP1 and phyllocladane (16*R*)-16,17-dihydroxyphyllocladan-3-one) have been reported to have no effect on motor coordination [20,31]. In a similar outcome, NPT lacked locomotor coordination impairment and sedative effects in the rotarod and pentobarbital-induced sleeping tests, respectively. This is important because the anxiolytic-like activity of NPT is not attributed to its sedative properties like benzodiazepines [17]. 

## 4. Materials and Methods

### 4.1. Drugs

Neophytadiene was obtained from Santa Cruz Biotechnology (Santa Cruz, CA, USA) and had a purity of at least 95% according to the manufacturer). Clonazepam (CNZ), pentylenetetrazole (PTZ), flumazenil, 4-aminopyridine, and bicuculline were acquired from Sigma-Aldrich (St. Louis, MO, USA). Pentobarbital was purchased from Pisa Farmaceutica (Mexico City, Mexico).

### 4.2. Animals

Male CD-1 mice, weighing 38 ± 2 g within 8–10 weeks old, were provided by the Institute of Neurobiology at the Autonomous University of Mexico in Juriquilla, Queretaro, Mexico. The experiments were conducted in the Bioterium at the University of Guanajuato and in the Behavioral Unit of the Neurobiology at UNAM. Animals were housed in groups of 5 per polycarbonate cage, maintained on a 12 h: 12 h light-dark schedule with free access to water and a standard diet (LabDiet 5001). After 1 week of adaptation, the mice were randomly distributed into groups of 8 animals and used only once. All tests were conducted between 08:00 a.m. and 02:00 p.m. Experimental protocols were approved by the Ethics Committee of the University of Guanajuato (CIBIUG-P03-2020) before the beginning of all experiments. Behavioral assessments were conducted in quiet areas by observers unaware of the treatment administered.

### 4.3. Pharmacological Treatment 

Neophytadiene (NPT) and all drugs were prepared in 0.5% (*w*/*v*) carboxymethyl cellulose (CMC)-physiological saline solution and administered orally (p.o.). Each experimental group consisted of a total of eight mice randomly distributed. The treatments were administered 1 h before each evaluation. Groups of mice were treated with NPT (0.1, 1, and 10 mg/kg) or positive controls 1.5 mg/kg clonazepam (CNZ) and 20 mg/kg fluoxetine (FLX) [32]. The doses of NPT were selected considering the percentage present in *Blumea lacera* leaves [3] and the results of preliminary studies carried out in our laboratory.

### 4.4. Light–Dark Box Test (LDBT)

The dimensions of the light–dark box were one-third of the dark compartment (illumination of 40 lux) and two-thirds of the light compartment (illumination of 400 lux) with an exterior size of 46 × 27 × 30 cm (l × b × h). Each mouse was placed into the dark section, and the crossing between the two sections and the time spent in the light section were recorded for 5 min [33]. The parameters evaluated in the light section were time in the compartment and the number of entries. 

### 4.5. Elevated Plus-Maze Test (EPMT) 

The methodology and apparatus used were those described by Lister [34]. The test began with the placement of a mouse in the center of the elevated plus-maze, positioned in front of the closed arm. The animal was allowed to roam freely for 5 min, and the entire trial was recorded. The parameters determined were the number of entries in the open and closed arms and the time spent in the open and closed arms. The increase in time and number of entries in the open arms were indicative of anxiolytic-like activity.

### 4.6. Open Field Test (OFT)

A mouse was settled in the center of a 42 × 42 × 42 polyvinyl chloride box. The movement around the central and peripherical areas of the box was recorded with a camera [35]. The total distance, the resting time, the time in central squares, and the distance in center squares were the parameters evaluated for 5 min.

### 4.7. Hole–Board Test (HBT)

The test consisted of a wooden box (42 × 42 × 30 cm) with 4 equidistant holes 3 cm in diameter in the floor. The test began with the placement of a mouse in the center of the hole-board. The mouse was allowed to explore the hole-board and the total number of head-dipping was counted for 5 min [36]. The anxiolytic-like mechanism of action of NPT was determined using 1 mg/kg bicuculline (BIC) (GABA_A_ antagonist) and 2 mg/kg flumazenil (FMZ) (GABA_A_ antagonist). These antagonists were administered intraperitoneally 15 min before the administration of the dose of 10 mg/kg NPT. The hole-board test was performed after 45 min of NPT administration. The data collected were total distance, resting time, time in central squares, and the distance in central squares.

### 4.8. Molecular Docking Study of Neophytadiene

The GABA_A_ receptor in complex with flumazenil was used in the docking study and it was obtained from the Protein Data Bank (www.wwpdb.org, accessed on 10 February 2023) [37] with the following accession code: 6D6T with the resolution of 3.86 Å. The molecular structure of neophytadiene (PubChem id: 10446) was constructed in MOE 2022.02 [38]. The MOE 2022.02 software was used to protonate the GABA_A_ receptor and neophytadiene structures. The molecular docking of neophytadiene was performed using AutoDock Vina 1.2.0 [39]. The grid box was fixed at the following coordinates: X = 119.613, Y = 169.091, and Z = 154.264 (centered covering the flumazenil coordinates), and the proportions of the grid were 25 × 20 × 20 points separated by 1 Å. We explored the possibility of binding with random molecular docking, centering at the following coordinates and grid dimensions: X = 127.392, Y = 167.341, and Z = 127.574, and grid size of 80 × 70 × 80 points. The pose with the lowest energy of binding was extracted for further analysis. We used the PyMOL software (The PyMOL Molecular Graphics System, Version 2.0 Schrödinger, LLC.) to visualize the complex structures.

### 4.9. Convulsion Test

Mice were injected with PTZ (90 mg/kg i.p.) or 4-Aminopyridine (4-AP) (12 mg/kg i.p.) (prepared in 0.9% saline solution). After 1 h, each animal received NPT or CNZ orally and individually placed in acrylic cylinders (23 × 27 × 6.5 cm). The onset and the duration of tonic–clonic convulsions and mortality of the mice were recorded [40]. All experiments were carried out between 10:00 a.m. and 02:00 p.m. For an additional test, mice were pretreated with 2 mg/kg i.p. flumazenil (FMZ) (antagonist of the GABA_A_ receptor) [41] 15 min before the administration of NPT 10 mg/kg and 45 min after the administration of 90 mg/kg PTZ.

### 4.10. Tail Suspension Test (TST)

The procedure consisted of wrapping adhesive tape around the mice’s tails, placing it 1 cm from the tip of the tail, and the suspension of every mouse 50 cm above the floor on the table’s edge. The immobility time(s) was evaluated for 6 min [42].

### 4.11. Pentobarbital-Induced Sleeping Test

The mice were administered a single dose of NPT (0.1, 1, 10 mg/kg) orally and 60 min after, the sodic pentobarbital (50 mg/kg) was injected intraperitoneally. Each mouse was observed for the onset and duration of sleep [43].

### 4.12. Rotarod Test

Before carrying out the test, the mice were individually placed in the rotarod apparatus (Harvard Apparatus, Barcelona, Spain) for 3 consecutive days so that they could adapt. The animals included in the test were those that managed to walk for 4 minutes at 4 rpm on the device [44]. The motor coordination test was performed at 60 and 120 min after the oral administration of the different doses of NPT. The mice were placed on the rotarod at the same speed (4 rpm) and the time spent on the apparatus was recorded [18]. The duration of the test was 4 min. The decrease in the time on the rotarod is indicative of impaired motor coordination.

### 4.13. Statistical Analysis of Data

All the data were reported as mean ± standard error of the mean (SEM) and analyzed by one-way or two-way analysis of variance followed by Dunnett’s multiple comparison test and Tukey’s multiple comparisons test, respectively. The statistical program used was GraphPad Prism (version 8.0.1). A probability value of *p* < 0.05 was considered a statistical difference.

## 5. Conclusions

The present investigation revealed that NPT has anxiolytic-like activity and anticonvulsant effects without sedative-locomotor effects in short-term studies. The experimental outcomes and the molecular docking analysis propose the possible participation of the GABAergic system in the anxiolytic-like and anticonvulsant activity provided by neophytadiene. The neuropharmacological actions of NPT will be assessed in subacute and chronic administration in murine models of Parkinson´s disease and other neurodegenerative diseases. The combination of NPT with other anxiolytic drugs will be assessed using isobologram studies.

## Figures and Tables

**Figure 1 molecules-28-03457-f001:**
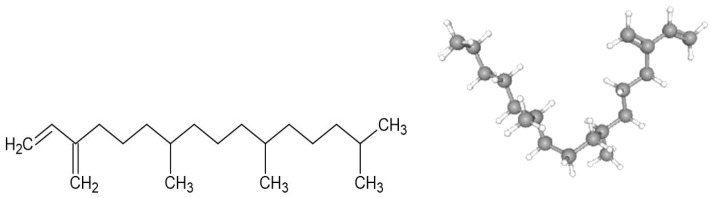
Chemical structure of neophytadiene shown in 2D structure and 3D conformer (NPT).

**Figure 2 molecules-28-03457-f002:**
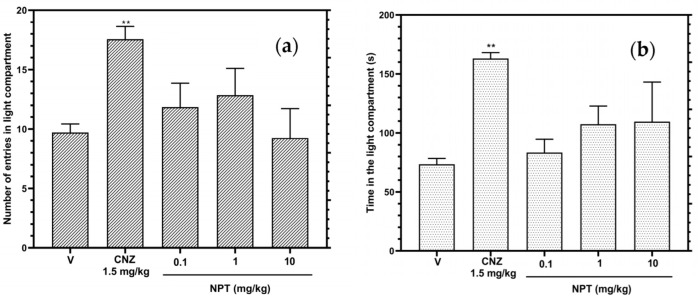
Anxiolytic-like actions of NPT (0.1–10 mg/kg p.o.) in the LDBT were determined by the time in the light compartment (**a**) and the number of crossings into the light compartment (**b**). The reference drug was 1.5 mg/kg p.o. clonazepam (CNZ). Bars represent mean values (±SEM) for the experimental group. *n* = 8, ** *p* < 0.05 compared to the vehicle group (indicated as V).

**Figure 3 molecules-28-03457-f003:**
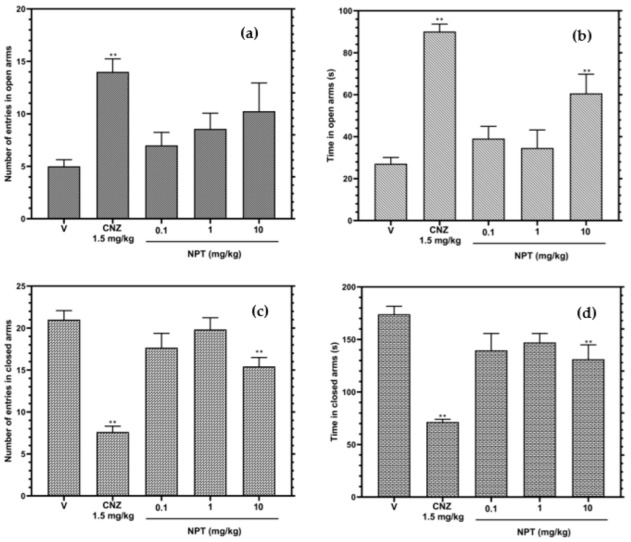
Anxiolytic-like effects of NPT (0.1–10 mg/kg p.o.) were assessed by the number of admissions in open (**a**) and closed arms (**c**) and the time elapsed in open (**b**) and closed (**d**) arms in the EPMT for 5 min. The reference drug was 1.5 mg/kg p.o. clonazepam (CNZ). The results of all experimental groups are reported as the mean values (±SEM). *n* = 8, ** *p* < 0.05 regarding to the vehicle group (indicated as V).

**Figure 4 molecules-28-03457-f004:**
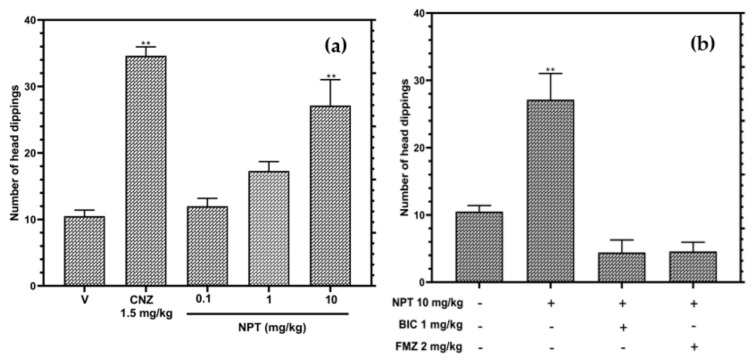
Anxiolytic-like actions of NPT (0.1–10 mg/kg p.o.) in the HBT (**a**), and the possible anxiolytic mechanism of NPT (**b**). Bicuculline (BIC) and flumazenil (FMZ) were used as antagonists. The reference drug was 1.5 mg/kg p.o. clonazepam (CNZ). The results of all experimental groups are reported as the mean values (±SEM). *n* = 8, ** *p* < 0.05 regarding to the vehicle group (indicated as V).

**Figure 5 molecules-28-03457-f005:**
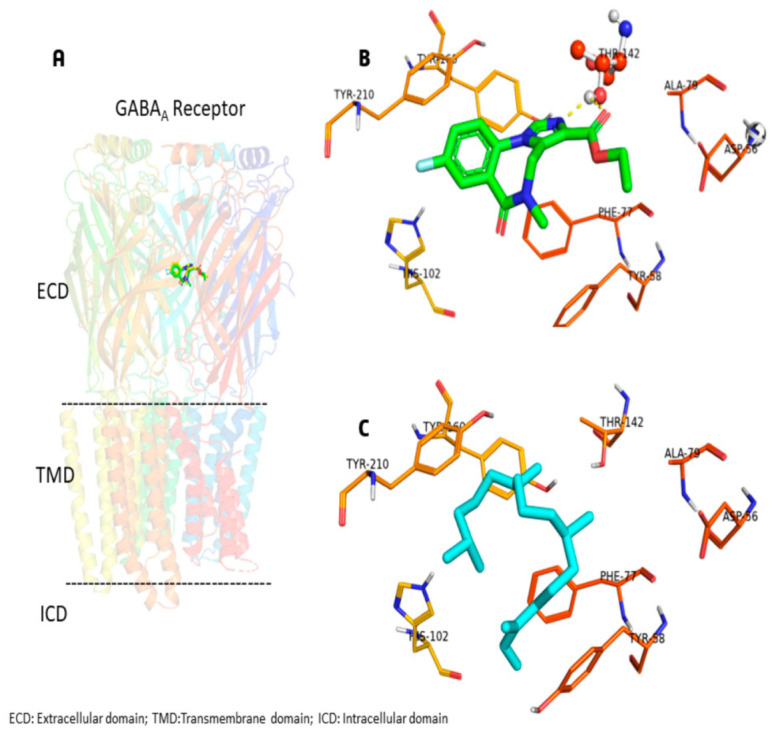
(**A**) Heteropentameric GABA_A_ receptor (PDB id: 6D6T) displayed in cartoon representation with competitive antagonist flumazenil (green color). (**B**) Flumazenil interacting with the binding pocket with hydrophobic residues, H-bond is shown in dashed line, and Thr142 is represented in ball and stick. (**C**) Neophytadiene (cyan color) is surrounded by hydrophobic residues in the pocket.

**Figure 6 molecules-28-03457-f006:**
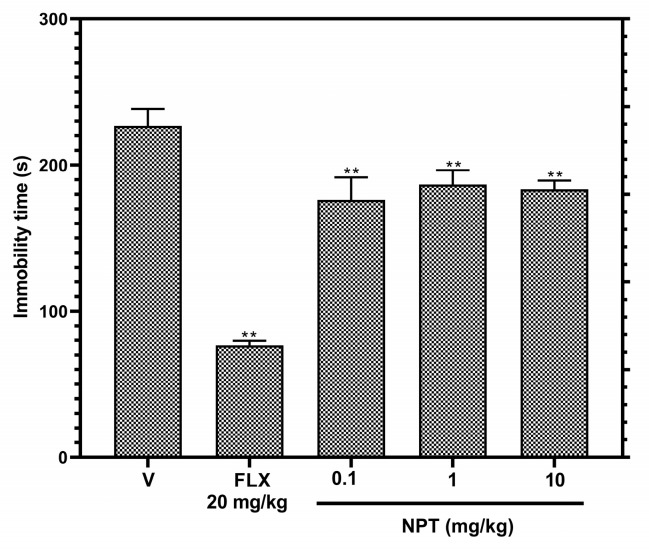
Antidepressant-like effects of NPT. Effects of NPT (0.1, 1 and 10 mg/kg p.o.) on immobility time were determined in the tail suspension proof. Fluoxetine (FLX) was used as the positive control. Bars represent mean values (±SEM) for the experimental group. *n* = 8, ** *p* < 0.05 compared to the vehicle group (indicated as V).

**Figure 7 molecules-28-03457-f007:**
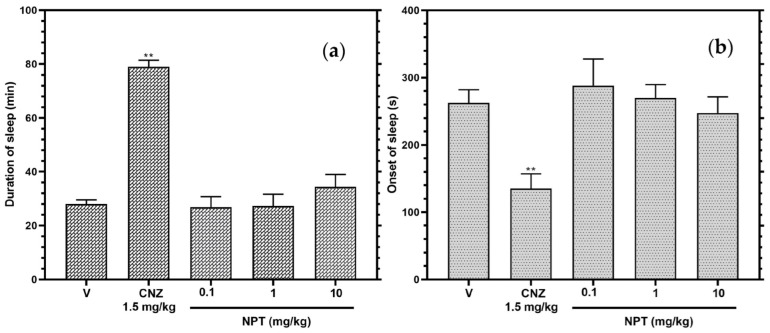
Sedative effects of NPT. The action of NPT (0.1, 1 and 10 mg/kg p.o.) was determined by the beginning of sleep (**a**) and length of sleep (**b**) in the pentobarbital-generated sleeping test. Clonazepam (CNZ) was the anxiolytic drug at 1.5 mg/kg p.o. The results of all experimental groups are reported as the mean values (±SEM). *n* = 8, ** *p* < 0.05 regarding to the vehicle group (indicated as V).

**Figure 8 molecules-28-03457-f008:**
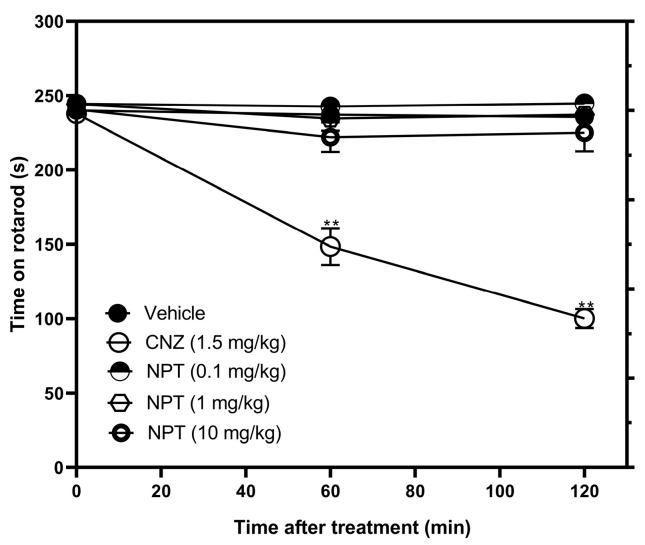
Locomotor actions of NPT. Effects of NPT (0.1–10 mg/kg p.o.) on time on rotarod were measured. Clonazepam (CNZ) was the anxiolytic drug at 1.5 mg/kg p.o. The results of all experimental groups are reported as the mean values (±SEM). *n* = 8, ** *p* < 0.05 regarding to the vehicle group (indicated as V).

**Table 1 molecules-28-03457-t001:** Effects of NPT in the OFT.

Treatment	Total Distance (cm)	Resting Time (s)	Time in Center Squares (s)	Distance in Center Squares (cm)
Vehicle	2070 ± 58.69	69.59 ± 2.45	16.08 ± 0.53	345.6 ± 13.85
CNZ 1.5 mg/kg	640.5 ± 16.57 ***	170.0 ± 4.15 ***	56.28 ± 1.54 ***	748.3 ± 12.59 ***
NPT 0.1 mg/kg	2081 ± 130.7	70.95 ± 11.56	19.94 ± 3.30	352.9 ± 55.97
NPT 1 mg/kg	2160 ± 295.6	73.99 ± 14.95	19.63 ± 3.40	348.5 ± 57.42
NPT 10 mg/kg	2252 ± 312.8	93.86 ± 15.71	14.53 ± 4.14	367.6 ± 81.51

CNZ-Clonazepam, NPT-neophytadiene. *** *p* < 0.0001 vs. vehicle group.

**Table 2 molecules-28-03457-t002:** Anticonvulsant activity of NPT in mice treated with 90 mg/kg pentylenetetrazole (PTZ) or 12 mg/kg 4-aminipyridine (4-AP) to induce convulsions.

Treatment	Onset of Convulsion (s)	Duration of Convulsion (s)	Mortality (%)
90 mg/kg PTZ			
Vehicle	69.34 ± 2.11	165.6 ± 3.77	100.00
CNZ 1.5 mg/kg	0.00 ± 0.00 ***	0.00 ± 0.00 ***	0.00
NPT 0.1 mg/kg	109.10 ± 13.12	47.71 ± 6.58 ***	14.00
NPT 1 mg/kg	97.86 ± 7.01	36.43 ± 6.61 ***	57.14
NPT 10 mg/kg ¥	73.43 ± 23.73	21.29 ± 6.50 ***	42.85
NPT 10 mg/kg + Flumazenil 2 mg/kg	100.9 ± 5.50	13.43 ± 2.88 ***	85.71
12 mg/kg 4-AP			
Vehicle	129.1 ± 6.95	38.63 ± 2.54	100.00
CNZ 1.5 mg/kg	258.1 ± 11.93 ***	15.50 ± 1.91 ***	50.00
NPT 0.1 mg/kg	176.9 ± 32.51	14.00 ± 2.78 ***	85.71
NPT 1 mg/kg	162.5 ± 9.74	18.88 ± 2.41 ***	87.50
NPT 10 mg/kg	191.3 ± 14.96 *	32.38 ± 4.79	75.00

CNZ-Clonazepam, NPT-neophytadiene. *** *p*< 0.0001 and * *p* < 0.05 vs. vehicle group. ¥ Two mice did not convulse.

## Data Availability

Data is contained within the article.

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
