# Peer review of "In Vivo Neuropharmacological Effects of Neophytadiene"

_molecules, 2023, doi:10.3390/molecules28083457_

Round 1

Reviewer 1 Report

The manuscript must be corrected as indicated.

Reviewer 2 Report

Castro and his co-worker reported the "In vivo Neuropharmacological Effects of Neophytadiene" in which they reveal that the NPT has anxiolytic-like activity and anticonvulsant effects without sedative-locomotor effects. By using molecular docking studies they determined the mechanism of neuropharmacological actions using inhibitors like flumazenil and analyzed the interaction of neophytadiene with the GABA receptor which is another addition to in vivo studies and gives a good understanding to readers. Whereas the author should conclude the future direction of the current work in the conclusion. I recommend manuscripts to publish in the given format.

Reviewer 3 Report

The manuscript presented In vivo study on the anxiolytic, antidepressant, anticonvulsant and sedative effects of neophytadiene using mouse models. While the study reported interesting observations, more convincing data and conclusions are required to warrant a good publication.

1.      A more scientific rendering of the chemical structure of neophytadiene is recommended.

2.      Number of mice need to be specified in each experiment.

3.  Should neophytadiene exert its pharmacological effect through its interaction with GABAA receptor, would flumazenil have similar effect given it has tighter binding affinity?

4.      Benzodiazepine family have been replaced by more effective compounds with fewer side effects. The less effective neophytadiene with reference to benzodiazepine as demonstrated herein does not lend credence to further investigation on neophytadiene as an anxiolytic candidate.
